# A domain in human EXOG converts apoptotic endonuclease to DNA-repair exonuclease

Michal R. Szymanski[1,2], Wangsheng Yu[1,2], Aleksandra M. Gmyrek[3], Mark A. White[2,3], Ian J. Molineux[4], J. Ching Lee[2,3] & Y. Whitney Yin[1,2]

Human EXOG (hEXOG) is a 5'-exonuclease that is crucial for mitochondrial DNA repair; the enzyme belongs to a nonspecific nuclease family that includes the apoptotic endonuclease EndoG. Here we report biochemical and structural studies of hEXOG, including structures in its apo form and in a complex with DNA at 1.81 and 1.85 Å resolution, respectively. A Wing domain, absent in other $\beta\beta\alpha$-Me members, suppresses endonuclease activity, but confers on hEXOG a strong 5'-dsDNA exonuclease activity that precisely excises a dinucleotide using an intrinsic 'tape-measure'. The symmetrical apo hEXOG homodimer becomes asymmetrical upon binding to DNA, providing a structural basis for how substrate DNA bound to one active site allosterically regulates the activity of the other. These properties of hEXOG suggest a pathway for mitochondrial BER that provides an optimal substrate for subsequent gap-filling synthesis by DNA polymerase $\gamma$.

[1] Department of Pharmacology and Toxicology, University of Texas Medical Branch, Galveston, Texas 77555, USA. [2] Sealy Center for Structural Biology, University of Texas Medical Branch, Galveston, Texas 77555, USA. [3] Department of Biochemistry and Molecular Biology, University of Texas Medical Branch, Galveston, Texas 77555, USA. [4] Department of Molecular Biosciences, University of Texas at Austin, Austin, Texas 78712, USA. Correspondence and requests for materials should be addressed to Y.W.Y. (email: ywyin@utmb.edu).

High concentrations of reactive oxygen species in the mitochondria result in a high incidence of mitochondrial DNA (mtDNA) oxidative damage, estimated to be about ten times greater than in nuclear DNA[1,2]. Human mtDNA encodes a subset of components for the oxidative phosphorylation electron chain that utilizes a proton potential to synthesize ATP. If left unrepaired, lesions on mtDNA can stall DNA replication and/or cause mutations, reducing mitochondrial function and cellular energy supplies[3]. Oxidative mtDNA damage has been attributed to increased cancer incidence and premature aging[4,5].

Base excision repair (BER) is the major mechanism for correcting mtDNA oxidative damage[6]. In this multi-enzyme reaction pathway, the DNA product from the previous reaction is the substrate for the next[7]. DNA intermediate 'hand-off' between the repair enzymes is therefore essential, as a prematurely released repair intermediate could be more detrimental than the original damage[8]. Increasing evidence indicates that the repair enzymes directly interact, forming a 'repairosome'[9].

Human mitochondrial BER is distinct from nuclear BER[10]. There is no designated repair polymerase in the mitochondria, and the replicase, Pol $\gamma$, is essential for both functions[11,12]. However, Pol $\gamma$ is least efficient in synthesizing DNA at a single-nucleotide gap and it lacks robust strand-displacement ability, activities necessary, respectively, for canonical nuclear single-nucleotide and long-patch (LP) BER[13]. However, the efficiency of Pol $\gamma$ gap-filling increases markedly on even a two-nucleotide gap[13]. If a nuclease could generate a larger gap at a lesion site, mitochondrial BER could yield products similar to that found in nuclear LP-BER, albeit through a different mechanism.

Human mitochondrial nuclease EXOG (hEXOG) is important for mtDNA integrity. hEXOG depletion selectively increases lesions in mtDNA, but has no effect on chromosomal DNA. Conversely, depletion of nucleases that are critical for nuclear BER, flap endonuclease 1 (FEN1) and DNA2 has no or limited effect on mtDNA[14]. Ectopic expression of hEXOG increases resistance to oxidative stress of proliferating myoblasts[15], and depletion of EXOG increases oxidative consumption rate in primary neonatal rat ventricular cardiomyocytes[16]. These studies provide strong support to the idea that hEXOG is involved in mtDNA repair. Importantly, hEXOG is found in a complex with other mtDNA repair enzymes, APE1, Pol $\gamma$ and ligase III; their interaction is enhanced by oxidative stress, and is regulated by poly(ADP-ribose) polymerase-1[17].

Human EXOG shares sequence homology with members of $\beta\beta\alpha$-Me nuclease family[18], which contains nonspecific endonucleases. One member, EndoG, is the paralog of hEXOG[19]. The nonspecific activity of EndoG is perfectly suited for activities during apoptosis when it digests single-stranded DNA and double-stranded DNA (dsDNA), and RNA. However, nonspecific activity is incompatible with a precise exonuclease activity necessary to process DNA 5'-end in BER[19–21]. In addition to endonuclease activity, hEXOG also possesses 5'-exonuclease activity; relative to the endonucleases, hEXOG harbours a C-terminal extension that may contribute to the new function.

Here we present enzymatic and crystallographic studies of hEXOG. Our results provide a detailed molecular and structural mechanism for hEXOG's substrate specificity and enzymatic properties in BER. The catalytic Core domain of hEXOG is identical to EndoG, but its C-terminal extension forms a unique Wing domain that converts the enzyme from a nonspecific endonuclease into a DNA-geometry-specific exonuclease. Our studies suggest that mtDNA repair is performed by long-patch BER that provides an optimal substrate for Pol $\gamma$ when fulfilling its role in DNA repair.

## Results

**Refolding and purification of hEXOG.** hEXOG is a mitochondrial membrane protein[19], and the hEXOG construct used here lacks the N-terminal trans-membrane domain (2–41 aa) and a predicted unstructured region (42–58 aa) of the native protein in an attempt to improve solubility; the resulting hEXOG-$\Delta$N58 enzyme is defined here as wild type. However, our hEXOG construct did not produce soluble protein; hEXOG was therefore partially purified from inclusion bodies as completely denatured protein and then refolded. Success of folding was assessed by the emergence of secondary structure features in circular dichroism (CD) spectra (Supplementary Fig. 1a). Our optimized procedure consistently refolded >85% of the total protein found in the inclusion body, increasing the refolding yield by ~1,000-fold over that previously attained[19]. Refolded hEXOG was then further purified by gel filtration chromatography to >95% homogeneity. The elution profile indicates that the 36.3 kDa hEXOG is a dimer (Supplementary Fig. 1b). All the independent enzyme preparations used in this work were equally active in assays and crystallized under the same conditions.

**Functional characterization of hEXOG.** We examined hEXOG exonuclease activity on substrates relevant to BER, including a 20 bp dsDNA and a 1-nt gapped DNA (Fig. 1; Supplementary Table 1). One strand was labelled with either 5'-$^{32}$P (Fig. 1a) or 3'-fluorescein (3'-F) (Fig. 1c) to allow quantification of reaction products. For simplicity, the DNA is numbered from its 5'-end (Fig. 1c). On the 5'-$^{32}$P-lablled duplex (Fig. 1a), the reaction product is a dinucleotide, suggesting that hEXOG has 5'-exonuclease activity. When the same 20 bp duplex is labelled with 3'-F, a 3'-F 18-mer forms, confirming that the enzyme excises a dinucleotide from the 5'-end (Fig. 1c). This property distinguishes hEXOG from typical exonucleases, which cleave the phosphodiester bond between the 1st and 2nd nucleotide, yielding mononucleotides. On the BER intermediate containing a 1-nt gap, hEXOG also produces 5'-dinucleotide products (Fig. 1a). hEXOG incises duplex and gapped DNA with equal efficiency (Fig. 1b), suggesting that the enzyme has higher affinity towards a 5'-P. To test this hypothesis, we directly measured the affinity of EXOG for DNA with 5'-P and 5'-OH. The $K_d$ values of hEXOG to 5'-P and 5'-OH dsDNAs measured by fluorescence anisotropy are 6.2 ± 2 and 22.2 ± 5.1 nM, respectively, indicating that hEXOG binds to 5'-P-DNA greater than threefold stronger than the 5'-OH-DNA (Supplementary Fig. 1c).

Under conditions of excess substrate, hEXOG generates product in two distinct phases where the first cycle of excision is much faster than subsequent reactions. The biphasic reaction persists at enzyme concentrations (Fig. 1d) that are higher than the apparent $K_d$ for DNA binding (Supplementary Fig. 1c). The two reaction rates allow us to calculate the steady-state turnover rate. Under the conditions employed, $k_{off}$ is 0.00028 s$^{-1}$ (Fig. 1e). Thus, while the first cycle of reaction is completed in <10 s, the turnover rate for the second reaction is 1 h, suggesting very slow product release. The nucleolytic reaction catalysed by hEXOG therefore approximates to single-turnover, which effectively eliminates unnecessary nuclease activity that is potentially detrimental during DNA repair. The single-turnover nature of the hEXOG reaction also enables us to measure the amount of product formed in one reaction cycle. When various concentrations of hEXOG were mixed with excess substrate, the product yield was ~50% of the active sites (Supplementary Table 2).

To determine whether both monomers of EXOG dimer are capable of binding to DNA, we measured hEXOG–DNA binding using electrophoretic mobility shift assay, and found that hEXOG monomers bind to 90% of DNA at 1:1 stoichiometry

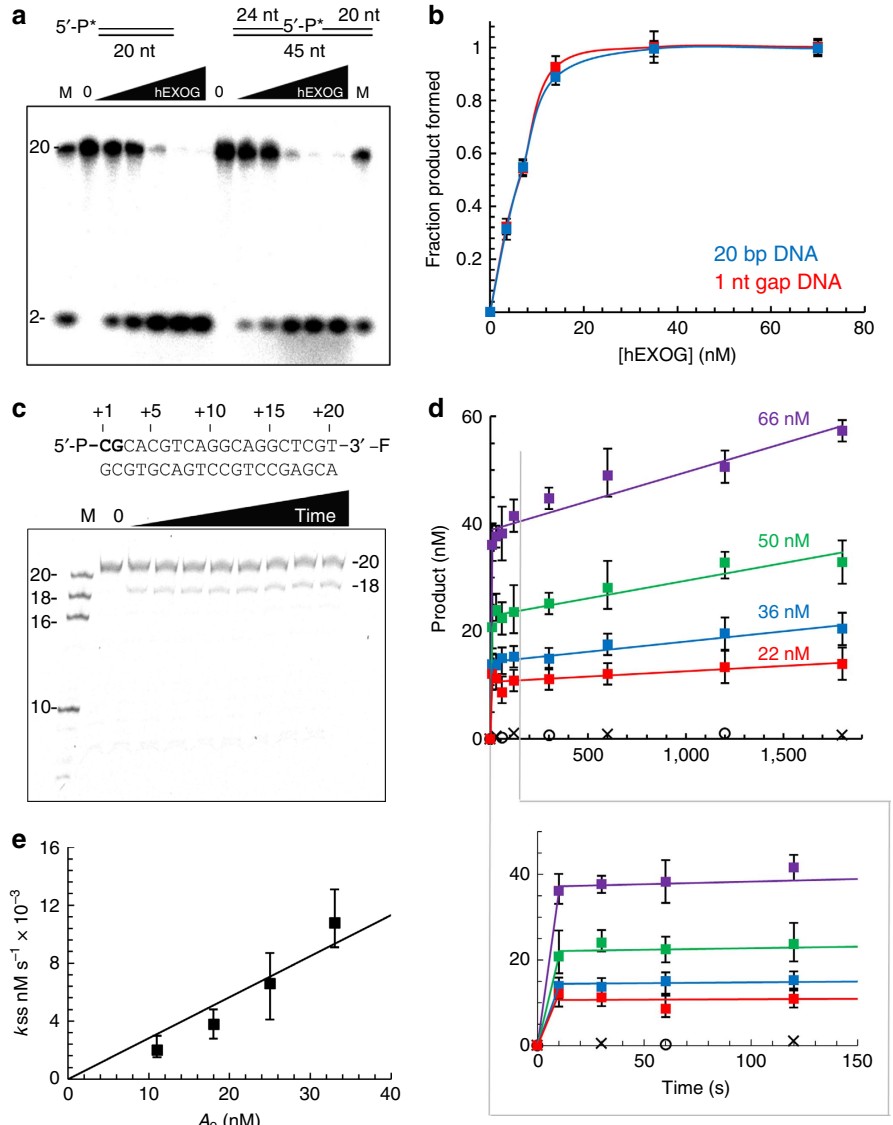

**Figure 1 | hEXOG exonuclease activity. (a)** Reactions used a 5'-$^{32}$P (*) 20 bp duplex or a 1-nt gapped DNA (10 nM) incubated with hEXOG$_{(monomer)}$ at 7, 14, 28, 70 and 140 nM for 5 min. Products were visualized after denaturing polyacrylamide gel electrophoresis. **(b)** Quantification of product from reactions in **a**. **(c)** Time-dependent exonuclease reaction using 200 nM 3'-F-labelled 20 bp duplex with 66 nM hEXOG$_{(monomer)}$. **(d)** Time-dependent product formation at various hEXOG$_{(monomer)}$ concentrations. Control reactions: H140A hEXOG (o) and wild-type enzyme + EDTA (×) are indicated in black. The initial rate of production formation is underestimated. **(e)** Plot of $k_{ss}$ versus $A_o$. Data represent the mean ± s.e.m. of three independent experiments.

(Supplementary Fig. 1d). The DNA-binding property of hEXOG is determined by isothermal titration calorimetry (ITC) using a nuclease-deficient (H140A) mutant. The resulting binding curve is bimodal; the simplest model that best describes the experimental data is with two distinct binding constants ($K_a$): $2.4 \times 10^5$ and $1.2 \times 10^7$ M$^{-1}$ (Supplementary Fig. 2), implying a different mode of binding for the second monomer of a dimer. The two affinities differ by 50-fold, suggesting that within certain DNA and enzyme concentrations, hEXOG binds only one DNA molecule, and the half-site reactivity is caused by the two different DNA-binding affinities in hEXOG dimer.

**Structure of homodimeric hEXOG in apo form.** To provide structural insights into the enzymatic properties of hEXOG, we first determined a crystal structure of hEXOG in its apo form at 1.81 Å resolution by molecular replacement using *Drosophila* EndoG (PDB code: 3ISM) as a search model[22]. The C-terminal extension (residues 300–369) of hEXOG that is absent in the

model was then built into a 2Fo-Fc electron density map. The final structure was refined to $R_{factor} = 16.1\%$ and $R_{free} = 18.7\%$ (Table 1).

Apo hEXOG is a symmetrical homodimer. Each monomer contains a Core domain that is extremely similar to that of EndoG (r.m.s.d. = 0.8 Å)[22–24], and a helical-bundle Wing domain that corresponds to the C-terminal extension (Fig. 2a,b). However, although the N-termini of the hEXOG and EndoG possess the same fold, they exhibit very different structures. The N terminus of each hEXOG monomer undergoes a domain-swap where the folding of the region is identical to that in EndoG, but the involved residues are from the opposing monomer rather than from the same monomer in EndoG: the residues $^{58}$K**AVL**EQ**FGFPL**T$^{70}$ (hydrophobic residues are bold) of one monomer collapse on a hydrophobic patch ($^{277}$**GLVFFP**HL$^{284}$) of the opposing monomer 30 Å away (Fig. 2a,d; Supplementary Fig. 3a). The swapped domains represent 45% of the total dimer interface in hEXOG, and thus contribute substantially to dimer

**Table 1 | Data collection and refinement statistics\*.**

| | apo hEXOG complex I | apo hEXOG complex II | hEXOG–DNA complex I | hEXOG–DNA complex II |
|---|---|---|---|---|
| PDB | 5T40 | 5T3V | 5T5C | 5T4I |
| hEXOG | Wild type | Wild type | H140A | H140A |
| Substrate DNA | — | — | 10 bp | 10 bp |
| Metal ion | $Mg^{2+}$ | $Mn^{2+}$ | $Mg^{2+}$ | $Mn^{2+}$ |
| Wavelength | 1.0 | 1.89 | 1.0 | 1.89 |
| *Data collection* | | | | |
| Space group | P $2_1$ | P $2_1$ | P $2_1 2_1 2_1$ | P $2_1 2_1 2_1$ |
| Cell dimensions | | | | |
| $a, b, c$ (Å) | 73.37, 83.73, 75.00 | 73.34, 83.44, 74.77 | 73.50, 80.27, 139.18 | 73.51, 79.72, 138.87 |
| $\alpha, \beta, \gamma$ (°) | 90.00, 113.54, 90.00 | 90, 113.42, 90.00 | 90, 90, 90 | 90, 90, 90 |
| Resolution (Å) | 34.7–1.81 (1.83–1.81)† | 49.7–2.6 (2.66–2.60) | 39.2–1.85 (1.87–1.85) | 40.0–2.39 (2.45–2.39) |
| $R_{sym}$ or $R_{merge}$ | 0.052 (0.814) | 0.058 (0.337) | 0.08 (1.0) | 0.114 (0.638) |
| $I/\sigma I$ | 25.8 (1.54) | 21.7 (2.3) | 31.5 (1.56) | 24.8 (2.4) |
| CC† | 1.000 (0.926) | 0.999 (0.989) | 1.000 (0.957) | 0.994 (0.994) |
| Completeness (%) | 98.10 (97.8) | 95.7 (93.3) | 99.9 (99.1) | 99.5 (97.4) |
| Redundancy | 3.6 (3.4) | 3.6 (3.6) | 7.2 (6.3) | 5.7 (4.4) |
| *Molecular replacement* | | | | |
| Model | 3ISM | 5T40 | 5T40 | 5T5C |
| LLG | 592.97 | 518.40 | 686.22 | 13,776.6 |
| TFZ | 18.2 | 42.6 | 34.6 | 57.2 |
| *Refinement* | | | | |
| Resolution (Å) | 34.7–1.81 | 49.7–2.6 | 39.23–1.85 | 40.03–2.39 |
| No. reflections | 74,273 | 24,370 | 70,421 | 32,854 |
| $R_{work}/R_{free}$ | 16.1/18.7 | 17.6/22.4 | 18.7/21.8 | 19.1/22.9 |
| No. of atoms | | | | |
| Protein | 9,552 | 9,552 | 9,465 | 9,464 |
| Ligand/ion | 22 | 12 | 1,148 | 1,144 |
| Water | 440 | 81 | 272 | 57 |
| *B*-factors | | | | |
| Protein | 39.37 | 63.54 | 58.41 | 63.16 |
| Ligand/ion | 57.95 | 127.37 | 32.09 | 57.31 |
| Water | 44.36 | 48.30 | 46.26 | 47.82 |
| r.m.s.d. | | | | |
| Bond lengths (Å) | 0.004 | 0.003 | 0.003 | 0.003 |
| Bond angles (°) | 0.7 | 0.483 | 0.59 | 0.53 |

hEXOG, human EXOG.
\*One crystal was used for each data set.
†Values in parentheses are for highest-resolution shell.

stability. A G227V substitution in the hydrophobic patch that disturbs the dimer interface of hEXOG displays markedly reduced activity[19], suggesting that dimerization is functionally important. A $Mg^{2+}$ is found in each active site, forming octahedron coordination with five water molecules and $N_\delta$ of N171. The nature of $Mg^{2+}$ was confirmed by $Mn^{2+}$ substitution and single-wavelength anomalous dispersion data (Supplementary Fig. 3b,c).

In contrast to the completely solvent exposed active site of EndoG, the active site of hEXOG is an enclosed V-shaped cleft. The cleft, measuring 32–35 Å in depth and 15 Å in width, is lined with positively charged residues from both Core and Wing domains, and is both electrostatically and dimensionally capable of accommodating dsDNA at its end (Fig. 2c). As the Core domain of hEXOG is highly similar to EndoG and other members of the ββα-Me superfamily that have no specificity, we hypothesized that the specific exonuclease activity of hEXOG would be attributed to the Wing domain. We therefore determined crystal structures of hEXOG complexed to substrate to understand substrate specificity and the half-site reactivity of the enzyme.

**Structure of hEXOG–dsDNA complex.** The nuclease-deficient variant, hEXOG–H140A, was refolded and purified similarly to

wild type (Supplementary Fig. 1) and used for co-crystallization with a 10 bp DNA. To distinguish the two strands of DNA in the duplex, only one strand contained a 5′-P. The structure was determined to 1.85 Å resolution by molecular replacement using the apo hEXOG structure. Ideal B-form dsDNA was built into the composite omit electron density map. The structure was then refined without any geometrical constraints on the DNA; the final structure has $R_{factor}$ and $R_{free}$ values of 18.7% and 21.8%, respectively (Table 1).

Successful crystallization experiments were performed at enzyme concentrations greatly exceeding the $K_d$ value for the weaker DNA-binding site of the hEXOG dimer as determined by ITC; consequently, DNA occupied both active sites (Fig. 3a). Both DNA duplexes are oriented such that the 5′-P end of the substrate is proximal and the 5′-OH end is distal to the active site. Thus, hEXOG has a clear preference for a 5′-P on the substrate strand. The 5′-OH containing strand thus defines the complementary strand.

**DNA binding disrupts the symmetry of the apo hEXOG dimer.** Binding to DNA does not change the structure of the Core domain; the r.m.s.d. between the apo and DNA complex forms is only 0.24 Å. However, the two Wing domains undergo different

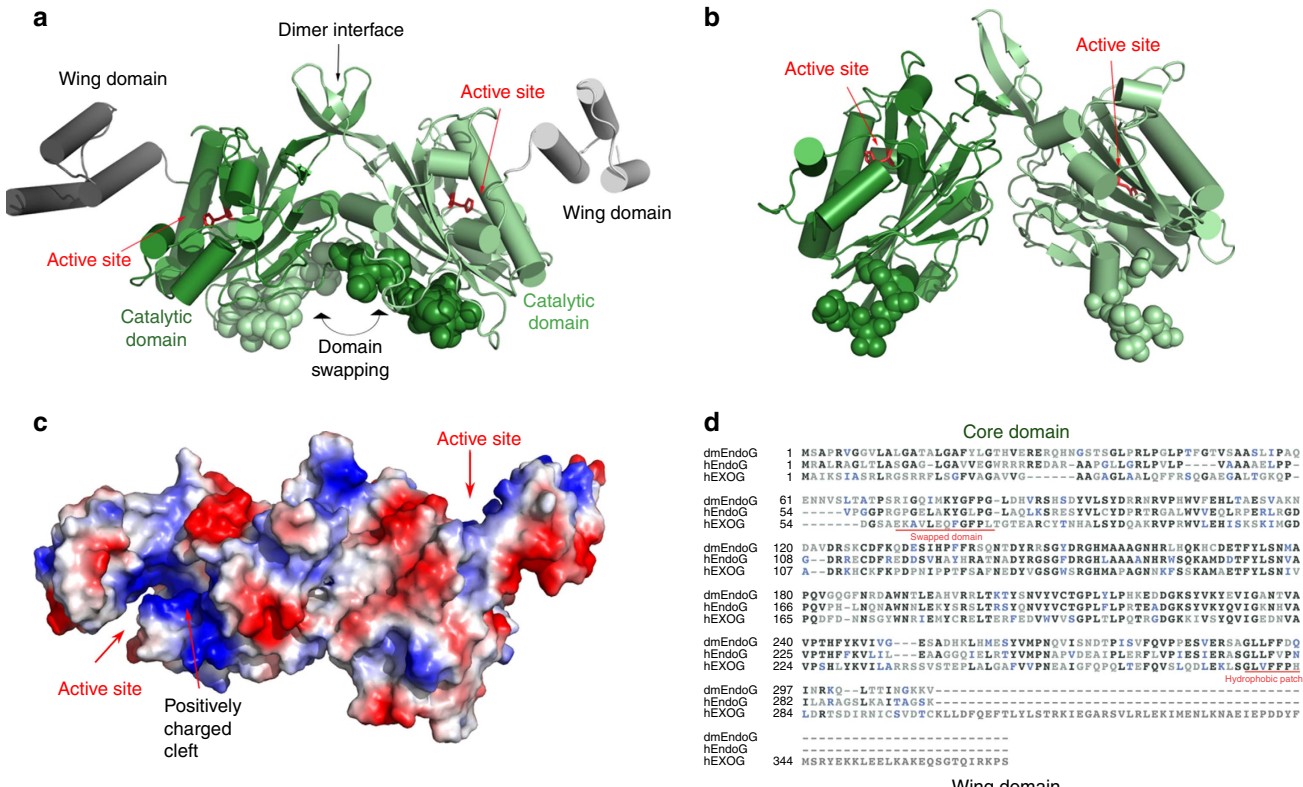

**Figure 2 | Structure of apo hEXOG.** (**a**) Structure of apo hEXOG. Each monomer has a Core (green) and a Wing domain (grey); residues involved in domain swapping are shown in CPK. (**b**) Structure of *Drosophila* EndoG (dmEndoG) with N-terminal residues shown in CPK. (**c**) Electrostatic surface of apo hEXOG. (**d**) Sequence alignment of hEXOG, hEndoG and dmEndoG.

conformational changes from their positions in the apo structure: one rotates only 3°, whereas the other rotates 35° away from the DNA (Fig. 3b,c,e). One active site therefore undergoes an atypical 'closed-to-open' conformational change. Consequently, upon DNA binding, hEXOG becomes asymmetrical where one active site remains in the 'closed' configuration and the other becomes 'open'. Thermal motion of the structure is revealed by changes in the B-factor. The average B-factors for the Wing domain of the two monomers in the Apo form are $\sim 75 \text{ Å}^2$ but in the DNA-bound form the B-factor of the open domain increases to $138 \text{ Å}^2$, whereas the closed domain remains at $76 \text{ Å}^2$ (Supplementary Fig. 4). These analyses suggest the 'open' Wing domain becomes less ordered than in the Apo structure, likely induced by the allosteric effect of DNA binding to the other active site.

**Substrate specificity of hEXOG.** The dsDNA is bound to each positively charged active site cleft; the substrate strand interacts primarily with the Core domain and the complementary strand with the Wing domain (Fig. 3a; Supplementary Fig. 5), explaining why the Wing domain of hEXOG increases specificity for dsDNA. Since neither Core domain of the dimeric hEXOG undergoes a structural change on binding DNA, the two substrate strands make identical electrostatic interactions (Fig. 3c,e). However, the conformational changes in the two Wing domains result in different DNA interactions. In the closed active site, the complementary strand interacts with Wing residues $R^{314}$, $R^{315}$, $R^{320}$, $R^{324}$ and $K^{327}$, with $R^{324}$ forming a bipartite interaction with the $+3$ nt DNA backbone at distances of 3.1 and 3.4 Å (Fig. 3e); $R^{315}$ and $K^{327}$ are, respectively, 4.3 and 6.7 Å from the $+2$ nt DNA backbone. In the open active site, the complementary DNA strand lacks several interactions, because the

bipartite distances to $R^{324}$ are increased threefold (to 9.4 and 9.8 Å), and $R^{315}$ and $K^{327}$ are similarly displaced (10.2 and 16.4 Å, respectively) from the $+2$ nt backbone (Fig. 3c). As electrostatic force is proportional to distance squared, a threefold increase in distance reduces the local attraction force to <10% of the 'closed' active site. These reduced interactions, together with the associated reduced desolvation of DNA, explain why the open active site has a lower affinity for DNA—and thus lower reactivity. These structural conclusions are in agreement with the two DNA-binding modes of hEXOG revealed by ITC.

To verify the structure of hEXOG free of a crystal lattice, we also determined the solution structure of hEXOG–DNA complex using small-angle X-ray scattering (SAXS). The SAXS hEXOG–DNA model established with CRYSOL[25] fits well with the crystal structure ($\chi^2 = 1.7$; Supplementary Fig. 6), indicating good agreement between solution and crystallographic structures.

**The tape measure for DNA cleavage.** The structural rigidity of the Core domain enabled us to generate a composite structure of an active substrate complex by replacing $A^{140}$ in the hEXOG–H140A:DNA complex with $H^{140}$ of the wild-type enzyme (Fig. 4a,b). In this complex, the 5'-P end of the substrate strand docks at the bottom of the active site cleft against residues from the Wing domain, forming clustered π–π interactions with $F^{307}$ and $Y^{310}$ (Fig. 4c). Two positively charged arginines, $R^{138}$ and $R^{314}$, bind to the DNA backbone at $+1$ and $+4$ nt, flanking the scissile bond (Fig. 4b). Interestingly, $R^{138}$ and $R^{314}$ bind two sulfate molecules in the apo structure in an essentially identical manner to the phosphdiester backbone of substrate DNA, reflecting their affinity for negative charges (Fig. 4a). Binding to DNA thus affords the substrate strand a register so that the oxygen bridging the $+2$ and $+3$ nt is placed in the catalytic

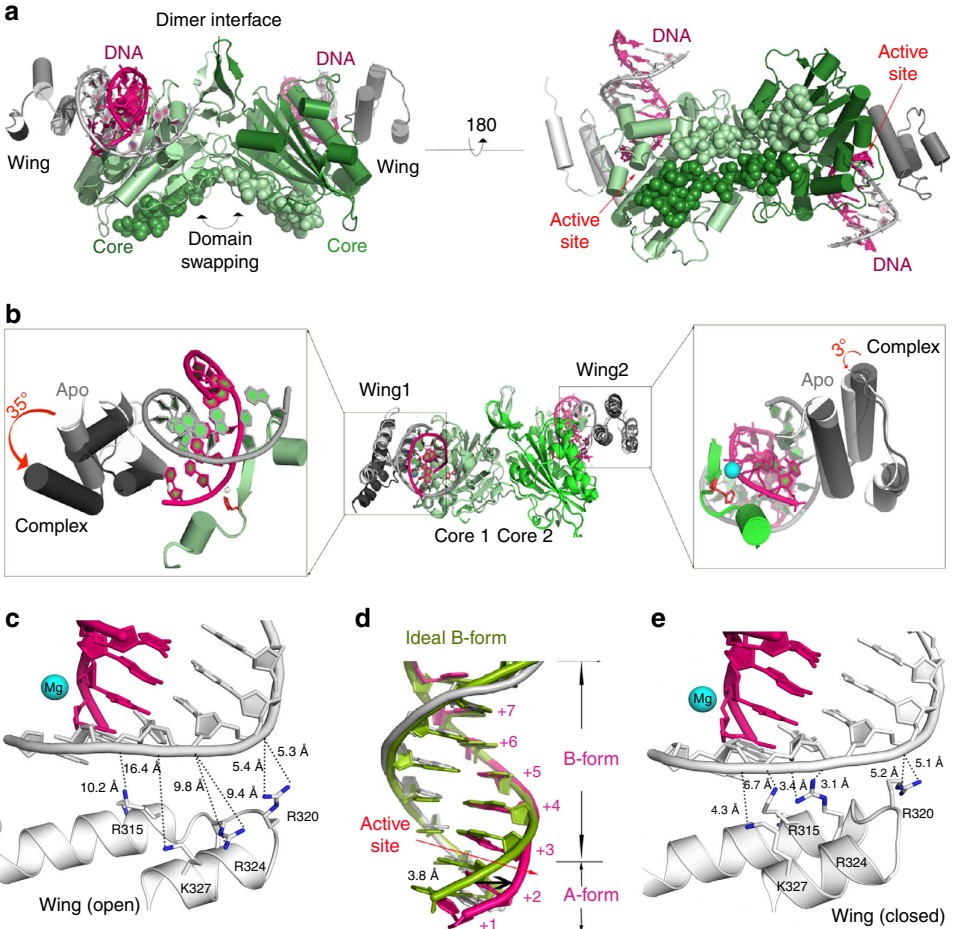

**Figure 3 | Structure of hEXOG–dsDNA complex.** (**a**) The complex has dsDNA in each active site. (**b**) Asymmetry of hEXOG upon DNA binding. Superposition of the apo enzyme and in a DNA complex in the open (left) and closed (right) monomer shows wing domain movements. (**c**) DNA interactions with the 'open' Wing domain. (**d**) The bound dsDNA (magenta and grey) partially transition to A-form from an ideal B-form (green). (**e**) DNA interactions with the 'closed' Wing domain.

active site. Because the Wing domain provides the register for DNA binding, it therefore serves as a 'tape-measure' for nucleolytic cleavage in a sequence-independent manner.

**Structural basis for catalysis.** In the substrate complex structure, DNA base pairs $+1$ to $+3$ adopt an A-form configuration with a 3′-endo sugar pucker (Fig. 3d; Supplementary Fig. 3d), while the remainder of the DNA is B-form and with 2′-endo or -exo sugar pucker (Supplementary Table 3). The B→A transition widens the minor groove, placing the substrate strand deeply into the active site cleft (Fig. 2) and bringing the O3′ of the $+3$ nt 3.8 Å closer to the $Mg^{2+}$ ion than if the DNA remained in B-form (Fig. 3d).

The $Mg^{2+}$ ion that was coordinated with five water molecules and the δ-N of $Asn^{171}$ in the apo structure is now coordinated with the bridging oxygen between the 2nd and 3rd nucleotide and the non-bridging oxygen $O_P$ of the $+2$ nt in the DNA complex (Fig. 4d). The $Mg^{2+}$ ion and $H^{140}$ are, respectively, 2.5 and 3.0 Å from the scissile bond, the distances that are competent for catalysis. The 3′-endo sugar pucker of the $+3$ nt selectively exposes the O3′ atom of the 2nd to 3rd phosphodiester linkage, facilitating nucleophilic attack at the scissile bond. While $H^{140}$ coordinates the attacking water, $Mg^{2+}$ serves as a Lewis acid that enhances the electrophilicity of the phosphorus so that it facilitates the nucleophilic reaction. The structure suggests that both $H^{140}$ and $Mg^{2+}$ perform catalytic roles (Fig. 4d). This conclusion is consistent with biochemical assays showing that the enzyme is

inactive in the absence of either $H^{140}$ or $Mg^{2+}$ (Fig. 1d). No potential attacking water molecules are observed in either the hEXOG–H140A–DNA complex or wild-type apo structures, suggesting that the attacking water requires stabilization by both the imidazole moiety of $H^{140}$ and the DNA backbone.

**Structural basis for slow product release.** In the substrate complex, the 5′-P of the substrate strand forms an electrostatic interaction with the positively charged $R^{314}$ (Fig. 5a). As hEXOG excises without fraying the DNA, the same interaction should persist in the product complex, where $R^{314}$ interacts with the 5′-P of the dinucleotide reaction product. We hypothesize that $R^{314}$ is, at least in part, responsible for both binding substrate and retention of the reaction product. Combining both properties into the same residue should reduce the rate of product release.

To test this structural prediction, we constructed the hEXOG mutant R314A that was refolded and purified similarly to wild type (Supplementary Fig. 1) and assayed its exonuclease activity. The apparent DNA-binding constant for the mutant is $35.7 \pm 7.2$ nM, that is, sixfold lower than wild type (Supplementary Fig. 1c), because eliminating the interaction between $R^{314}$ with 5′-P reduces substrate affinity. Notably, the reaction rate is at least twofold faster for the R314A mutant, together with its sixfold lowered binding affinity to the 5′-P-containing DNA (Fig. 5b,c), the increased catalysis is probably caused by faster product release and not increased chemistry, as $R^{314}$ is not near the active site.

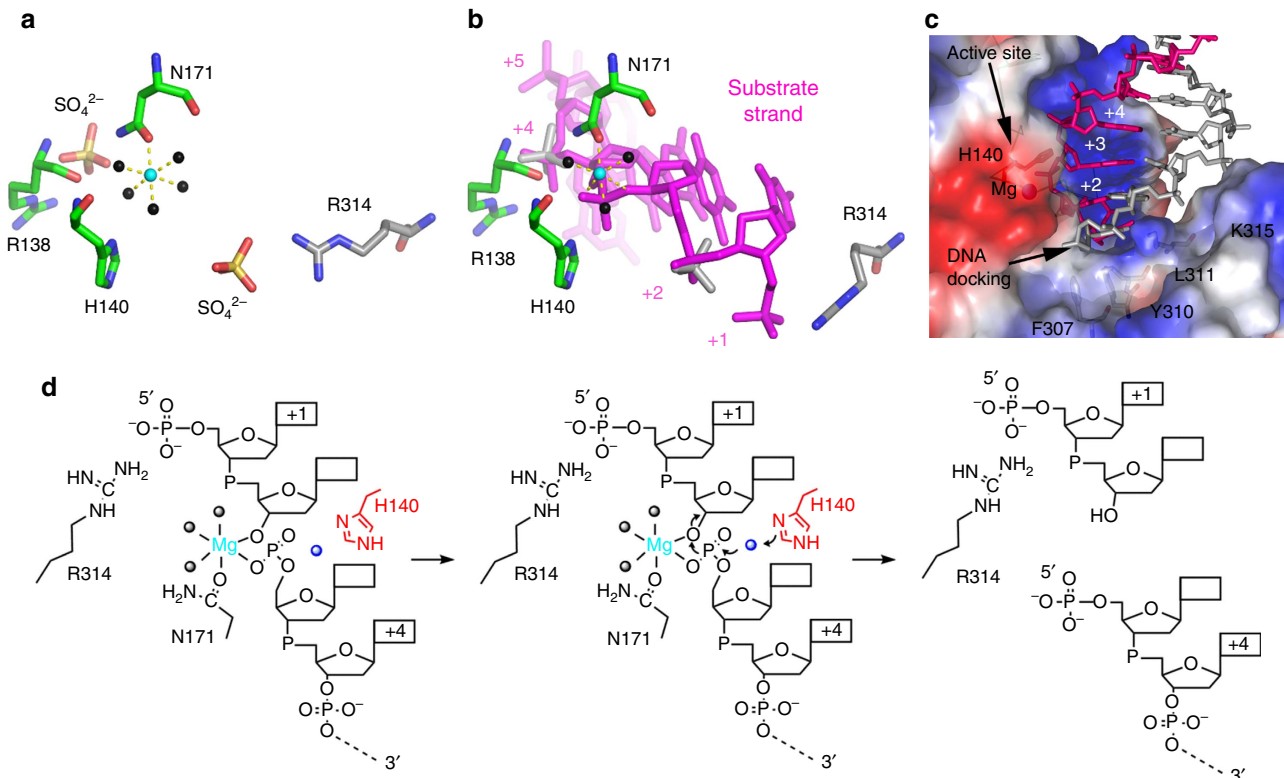

**Figure 4 | Proposed reaction mechanism for hEXOG.** (**a**) The active site of apo hEXOG illustrating $R^{138}$ and $R^{314}$ interactions with $SO_4^{2-}$ ions, and Mg (turquoise sphere) interactions with water molecules (black spheres) and N171. (**b**) The active site of hEXOG–DNA complex. DNA replaces water molecules in coordinating the $Mg^{2+}$ ion. $R^{138}$ and $R^{314}$ bind to the DNA backbone at the +1 and +4 nt, respectively. (**c**) The crystal structure of EXOG–DNA complex shows that the Wing domain forms a docking site for the 5′-P of the substrate strand. (**d**) Proposed mechanism of DNA cleavage by hEXOG.

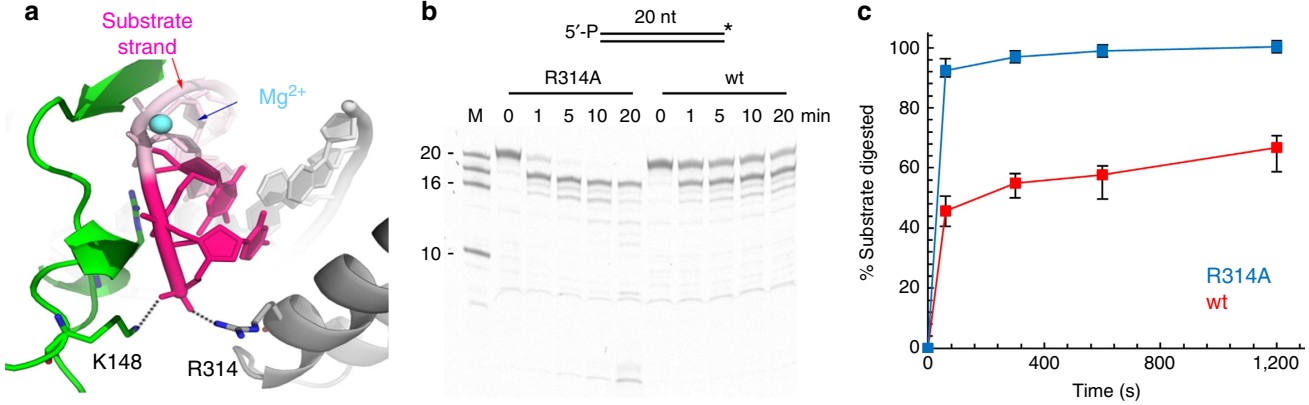

**Figure 5 | Structural basis for slow product release.** (**a**) Electrostatic interactions of $R^{314}$ and $K^{148}$ with the 5′-P of the substrate strand. (**b**) Time-dependent nucleolytic reaction of hEXOG R314A$_{(monomer)}$ (150 nM) on 3′-F dsDNA (200 nM). (**c**) Quantification of digestion. Data represent the mean ± s.e.m. of three independent experiments.

**Deletion of the Wing domain abolishes substrate specificity.** The structure of hEXOG–DNA complex suggests that the Wing domain is responsible for the tape-measured exonuclease activity and substrate specificity. To validate this conclusion, a Wing domain deletion mutant, hEXOG-ΔC68 was constructed. The mutant refolds such as wild type and remains a dimeric enzyme (Supplementary Fig. 1). When assayed on short DNA, the deletion mutant hEXOG-ΔC68 cleaves DNA randomly, as evidenced by the appearance of multiple product bands on a gel (Fig. 6a). Similar to the R314A mutant, hEXOG-ΔC68 also displays rapid turnover, providing further evidence that the Wing domain regulates nucleolytic activity. The hEXOG-ΔC68 mutation

eliminates critical residues involved in binding the 5′-P of both the substrate and product. The active site is completely open and the enzyme thereby loses both tape-measure controlled hydrolysis and any specificity for dsDNA (Fig. 6b). However, when assayed for endonuclease activity on circular φX174 DNA, hEXOG-ΔC68 exhibits catalytic activity that is eight to ten times higher than wild type (Fig. 6c,d). This increased activity of hEXOG-ΔC68 is most likely due to the loss of specificity for dsDNA, properties that should be shared by EndoG because hEXOG-ΔC68 and EndoG are similar in length and share the identical Core structure. The Wing domain of hEXOG therefore transforms a nonspecific nuclease into a substrate-specific exonuclease. Conversely,

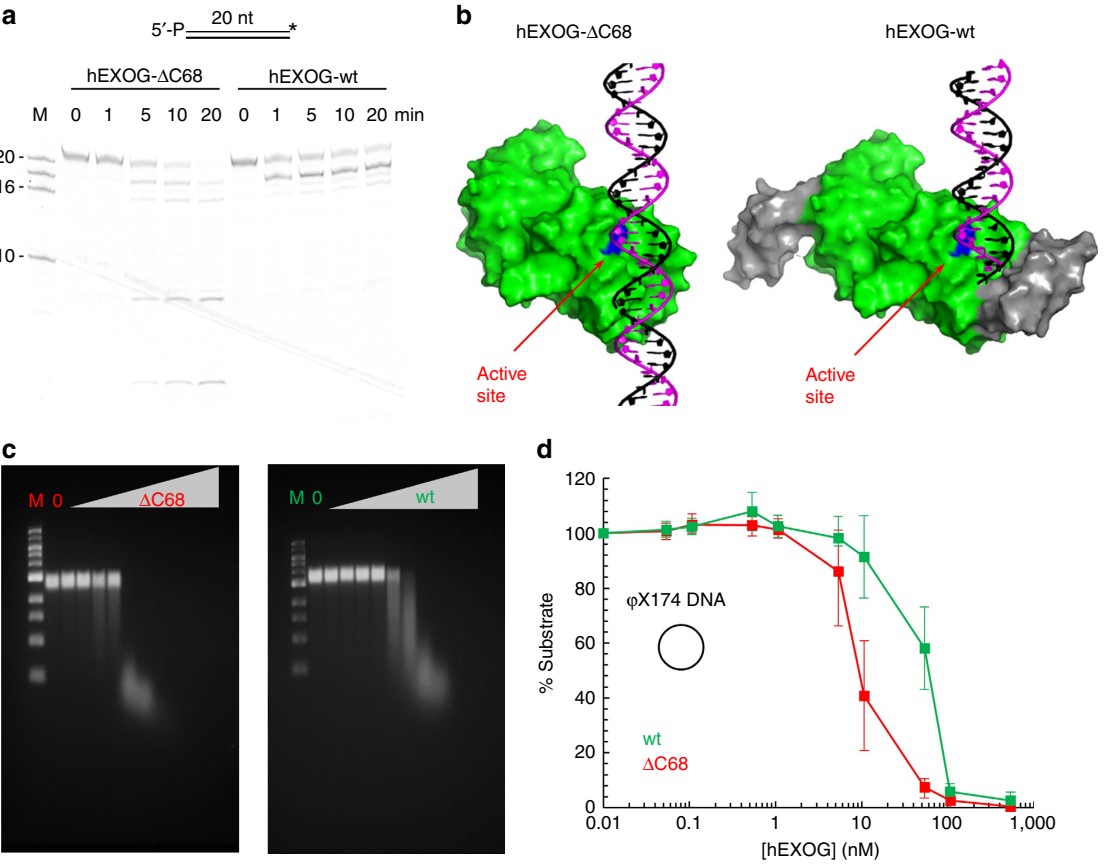

**Figure 6 | Wing domain deletion and substrate specificity.** (**a**) Digestion of 3′-F dsDNA (200 nM) by wild-type or ΔC68 hEXOG(monomrt) (150 nM). (**b**) Modelled hEXOG-ΔC68 showing loss of tape-measure and dsDNA specificity. (**c**) Endonuclease activity of hEXOG-ΔC68 and wild type on øX174 DNA (50 nM). (**d**) Quantification of **c**. Data represent the mean ± s.e.m. of three independent experiments.

by deleting the Wing domain from hEXOG, we have restored the basal nonspecific activity typical of the ββα-Me nuclease family.

## Discussion

The Wing domain provides hEXOG with features that are commonly seen in other DNA repair nucleases[26], including substrate specificity, slow product release and self-regulated nucleolytic reaction. Importantly, hEXOG enhances Pol γ DNA synthesis by providing an optimal substrate[13], therefore increasing the overall efficiency of the mtBER pathway. The Wing domain exerts its effect in several ways. First, it remodels the DNA-binding site of an EndoG-like enzyme to gain its preference for duplex DNA. Second, the Wing and Core domains each contribute a positively charged arginine, $R^{314}$ and $R^{138}$, in binding the substrate DNA backbone flanking the scissile bond (Fig. 7a). Third, $R^{314}$ of the Wing domain forms an electrostatic interaction with the substrate strand 5′-P, enhancing 5′ exonuclease activity and reducing product release. Fourth, the Wing domain provides a tape-measure by positioning the dsDNA substrate for an incision between the 2nd and 3rd nucleotides. Canonical BER can be halted under oxidative environment because oxidized 5′-phophosphate ribose by formation of a covalent bond with an active site lysine in the 5′-phophosphate ribose lyase[27]. Since the active site in hEXOG is distant from the 5′-end of the DNA, crosslinking of the 5′-end of the DNA substrate with hEXOG will not impact its nuclease activity. This mechanism of cleavage suggests that hEXOG is likely to process a damaged 5′-end regardless of chemical structure, circumventing a potential blockage in the DNA repair pathway.

In mitochondrial BER, Pol γ performs gap-filling synthesis following end-processing of the damaged DNA. Because Pol γ is ineffective in synthesizing at a single-nucleotide gap, only a nuclease that generates DNA products with a gap size greater 2 nt is suitable for the known activities of Pol γ[13]. Among the repair nucleases found in human mitochondria, FEN1 (ref. 28), helicase-nuclease Dna2 (refs 28,29) and hEXOG[14], only the latter provides this activity. Together with the ability of hEXOG to interact with Pol γ and DNA ligase III, this study has established a precise mechanism of hEXOG activity that makes it indispensable for mtDNA repair. A different mitochondrial nuclease, MGME1 (mitochondrial genome maintenance exonuclease 1) processes single-stranded and flap DNAs, but has low activity towards BER intermediates, gapped DNAs[30,31]. Hence, hEXOG and MGME1 divide the repair and replication functions in mtDNA maintenance.

The Wing domains become asymmetrical upon binding to DNA: one monomer interacts with the substrate tightly and the active site closes, the other active site remains open and flexible, and binds DNA loosely. This negative allosteric regulation is most likely mediated through the Core domains connecting action in the active site of one Wing domain to the other. In an attempt to map the connecting path between the two, we noticed that, although binding to DNA does not significantly change the stability of the Core domains, local thermal motion is increased in three important regions: the active site (residues 140–150), the $Mg^{2+}$-binding site (residues 170–190) and a portion of the dimer interface (residues 203–215) (Supplementary Fig. 4). The involvement of these regions needs further investigation but their locations are suggestive of a pathway allowing inter-domain allosteric regulation.

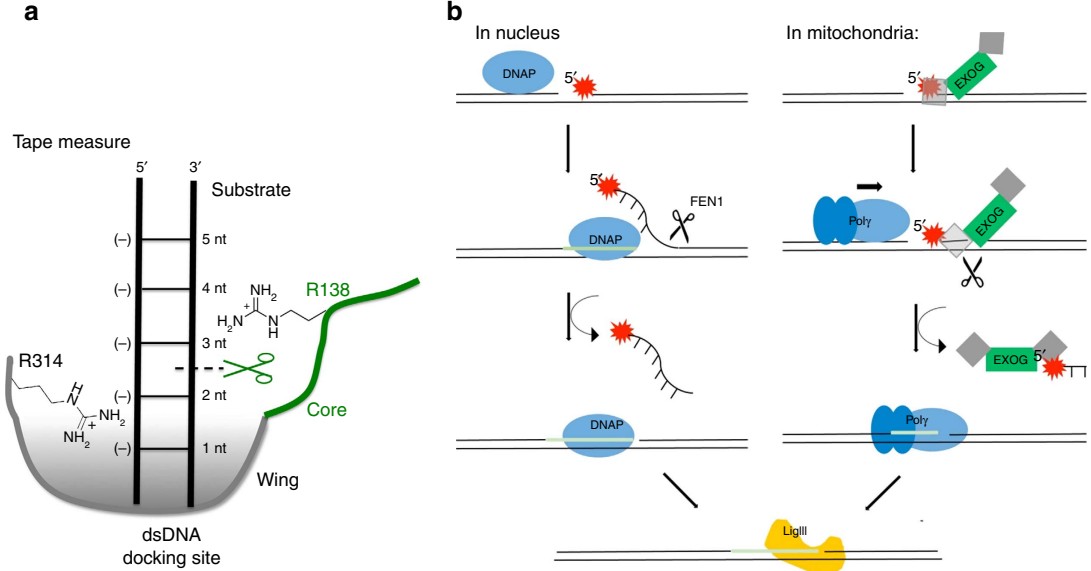

**Figure 7 | Graphic summary and a mitochondrial BER model. (a)** Structural basis for hEXOG activity. **(b)** A proposed model for mitochondrial DNA long-patch (LP) BER and its difference from the nuclear LP-BER. In mitochondria, EXOG exerts 5′-exonuclease activity at the abasic site and generates a 2-3 nt gap on dsDNA, which is an optimal substrate for Polγ to perform gap-filling synthesis. In contrast, nuclear LP-BER depends on the strand displacement activity of the DNA polymerase and FEN1 cleaves the DNA flap.

While allostery provides an explanation for the observed half-site reactivity, it leaves open the question why hEXOG is a homodimer, as only one active site is necessary for BER. A possible function for the second Wing domain is to interact with other repair enzymes. Protein–protein interactions are thought to enhance repair efficiency by assuring transfer of DNA intermediates between enzymes in the repair pathway and preventing inappropriate DNA transactions[8,9].

The activities of hEXOG that we have characterized strongly suggest that mtDNA repair is performed via a LP-BER pathway. However, the reactions in the pathway are different from nuclear LP-BER[10,32]. The two pathways diverge after the generation of a gapped DNA by APE1 (Fig. 7b). In nuclear LP-BER, DNA synthesis occurs before the nuclease reaction; Pol δ synthesizes DNA across the lesion site and displaces the downstream strand. FEN1 then removes the DNA flap before ligation. Conversely, in mitochondrial LP-BER, the nuclease reaction occurs before DNA synthesis: hEXOG processes the 5′-end of DNA downstream of the lesion, generating an optimal substrate for DNA synthesis catalysed by Pol γ. Mitochondrial BER thus successfully circumvents the lack of strand-displacement activity by Pol γ and its inefficiency in synthesizing at single-nucleotide gaps. This model also explains why FEN1 activity is dispensable in the mitochondria[14,33].

The structural and functional studies presented here provide a missing piece of the puzzle towards our understanding the mechanism of mitochondrial BER. The LP-BER pathway we propose allows DNA repair to proceed through a well-controlled 'cut-and-fill' mechanism that generates an optimal substrate for Pol γ without the need for strand-displacement and without requiring flap-endonuclease activity.

## Methods

**Cloning and purification of hEXOG.** hEXOG was cloned in pET22b without N-terminal residues 2–58 and with a C-terminal His-tag. This clone served as the parental construct for the H140A and R314A substitutions and for the C-terminal 68 aa deletion (hEXOG-ΔC68). All hEXOG variants were expressed in *Escherichia coli* ROSETTA (DE3) (Novagen). Cells were inoculated with 0.1% of an overnight culture in LB media containing ampicillin ($100 \mu g \, ml^{-1}$) and chloramphenicol ($34 \mu g \, ml^{-1}$) at 37 °C to $OD_{600} = 0.4$. Isopropyl-β-D-thiogalactoside (1 mM) was then added and the culture was incubated for 12 h at 25 °C. All proteins were purified using a modification of a previous protocol[34]. Briefly, the insoluble hEXOG

in inclusion bodies was solubilized in denaturing buffer (10 mM Tris, pH 8.2, 200 mM NaCl, 20 mM B-ME, 10 mM imidazole and 8 M urea). The unfolded protein was purified on a Ni-NTA affinity column. The eluted hEXOG was flash diluted in denaturing buffer to concentration of $1–2 \mu M$, and then refolded, first by dialysing against denaturing buffer where 8 M urea was substituted with 1.5 M sorbitol and then against GF buffer (20 mM HEPES, pH 8.0, 300 mM NaCl, 1 mM TCEP) with 5% glycerol (v/v). The refolded hEXOG was loaded onto a Hi-Load Superdex 200 gel filtration column (GE Healthcare Life Sciences), and peak fractions were concentrated using a Centricon concentrator (Millipore) to $\sim 7 \, mg \, ml^{-1}$. The enzyme used for activity assays was flash-frozen in GF buffer with 40% glycerol in liquid nitrogen and stored at $-80$ °C. Before the experiments, the protein was dialysed against buffer EK140 (20 mM HEPES (pH 7.5), 140 mM KCl, 1 mM TCEP and 5% (v/v) glycerol) and the concentration was determined spectrophotometrically using extinction coefficient $\varepsilon_{280(dimer)} = 77,030 \, M^{-1} \, cm^{-1}$, for wild type, R314A and H140A, and $\varepsilon_{280(dimer)} = 68,090 \, M^{-1} \, cm^{-1}$ for hEXOG-ΔC68.

**Circular dichroism spectroscopy.** Unfolded hEXOG–H140A ($\sim 2 \mu M$) was used as the baseline for monitoring folding; the protein was the Ni-NTA eluant dialysed against 20 mM $Na_2HPO_4$ (pH 7.5) containing 8 M urea to remove the imidazole. After testing various folding procedures, hEXOG wild type, H140A or ΔC68 variants ($2 \mu M$) was dialysed against 20 mM $Na_2HPO_4$ (pH 7.5) with 140 mM KCl overnight at 4 °C. CD spectra (195–260 nm) were collected on a Jasco J-815 CD Spectrometer at 20 °C. Each reported CD curve is an average of three scans, corrected for the contribution from the buffer alone.

**Small-angle X-ray scattering.** SAXS data were collected using a Rigaku BioSAXS-1000 with an FRE+ + Copper x-ray source. Exposure times were 8, 12 and 16 h at each apo hEXOG concentration, and 12, 14 and 16 h at each DNA-bound hEXOG concentrations, with matching times for their buffers. Hourly partial scans were visually examined for signs of radiation damage and the $\chi^2$-squared metric used to ascertain when damage began (http://xray.utmb.edu/saxns-XCS.html). Images were processed in SAXLab (Rigaku) and buffer subtraction performed with the SAXNS_ES web server (http://www.xray.utmb.edu/saxns-es.html). The apo hEXOG data could be merged from the 1.0 and 0.75 mg ml$^{-1}$ samples based on their XCS values. The hEXOG–DNA sample data from the 6, 4 and 2 mg ml$^{-1}$ data were merged, using cross-$\chi^2$ as a guide to exclude regions with problems due to near-beamstop parasitic scatter or a poor signal to noise ratio. All data were examined using PRIMUS, $P(r)$ curves generated with GNOM, and ten molecular bead models generated with DAMMIF then aligned and filtered with DAMAVER, all from the ATSAS suite[35]. Crystal structures were compared to the SAXS data using CRYSOL and rigid-body models refined using CORAL[35].

**Preparation of oligonucleotide substrates.** Oligonucleotide containing 3′-fluorescein (Midland Certified Reagent Co) (Supplementary Table 2) or 5′-$^{32}$P, prepared using T4 kinase and [γ-$^{32}$P-ATP], was annealed with the complementary strand by heating at 95 °C for 5 min in a solution containing 20 mM Tris, pH 8.1, 100 mM NaCl, 10% (v/v) glycerol and slowly cooled to room temperature.

**Isothermal titration calorimetry.** hEXOG–H140A and DNA were dialysed separately against EK140 buffer at 4 °C. Titrations were conducted using a VP-ITC calorimeter (Microcal, GE Healthcare) at 20 °C with 23 injections (1 μl for the initial injection, followed by five injections at 5 μl, eight injections at 10 and 15 μl for the subsequent injections) of 60 μM DNA titrant into 1.437 ml 2.8–3.2 μM hEXOG–H140A, with 360 s between injections. Reference titrations were performed without enzyme. The first injection was excluded from the analysis. Thermodynamic binding parameters were derived from fitting isotherm to two sequential binding sites model, using SEDPHAT[36].

**Endonuclease activity assay.** hEXOG wild type or mutants were incubated with circular ssDNA (100 ng μl$^{-1}$ ϕX174 virion DNA) in EK140 buffer for 10 min at 25 °C. Reactions were initiated by the addition of MgCl$_2$ to 5 mM, and incubated for 1 h at 25 °C using varying enzyme concentrations. Reactions were stopped by the addition of stop BF (50 mM EDTA, 0.1% SDS, 80% formamide). Cleavage products were resolved by electrophoresis in a 1% agarose gel, stained with ethidium bromide, and quantified using ImageJ (http://rsb.info.nih.gov/ij/). Substrate remaining was quantified with the initial concentration set at 100%.

**Exonuclease activity assay.** hEXOG wild type or a variant was pre-incubated with MgCl$_2$ (5 mM) for 10 min at room temperature in EK140 buffer. The reaction was initiated by the addition of DNA substrate; enzyme and DNA concentrations are given with the figures. All experiments were performed at 25 °C. Aliquots were taken and mixed with an equal volume of stop BF and incubated at 95 °C for 4 min. Reaction products were separated using denaturing 23% polyacrylamide/7 M urea polyacrylamide gel electrophoresis and visualized using a Molecular Dynamics Storm 860 Phosphorimager. Product bands were quantified using ImageQuant (GE Heathcare). The time dependence of product formation is fitted to the equation [product] = $A_o + k_{ss} t$, where $A_o$ is amplitude, $k_{ss}$ is steady-state turnover rate and $t$ is time. The burst product is approximated to $A_o$. The off rate $k_{off} = k_{ss}/A_o$. Reactions were stopped with stop BF and incubated at 95 °C for 4 min. Error bars represent ± s.d. from three parallel experiments.

**DNA-binding assays.** Substrate 10 bp DNA (10 nM) containing a 5′-P and a 3′-fluorescein (Supplementary Table 1) was mixed with increasing concentrations of hEXOG wild type or R314A mutant in EK140 buffer plus 20 mM EDTA. To determine the affinity of hEXOG to 5′-P and 5′-OH on DNA, the same 3′-F-labelled DNA was synthesized containing a phosphate or a hydroxyl group. The measurements were taken on a ISS PC1 photon counting spectrofluorimeter by monitoring the emission of the F-labelled nucleic acid ($\lambda_{ex} = 480$ nm and $\lambda_{em} = 520$ nm) as described previously and the experimental titrations were fit to single-site binding isotherm[37]. Electrophoretic mobility shift assay was performed with 10 nM 5′-$^{32}$P-labelled 10 bp DNA mixed with increasing concentrations of hEXOG wild type or R314A mutant in EK140 buffer plus 20 mM EDTA. The mixtures were resolved on a 10% acrylamide gel at 4 °C in × 0.5 TBE supplemented with 20 mM EDTA. Gels were scanned and quantified as above. The fraction of DNA bound was calculated after background correction. Apparent $K_d$ values were estimated by fitting using Mathematica.

**Crystallization and data collection.** Apo hEXOG crystals were obtained by the hanging-drop method at 19 °C using 5.6 mg ml$^{-1}$ enzyme with 5 mM MgCl$_2$ against a well solution containing 2.0 M (NH$_4$)$_2$SO$_4$, 0.1 M MgCl$_2$ and 0.1 M Tris-HCl pH 8.5. Crystals containing Mn$^{2+}$ were obtained by soaking hEXOG-Mg$^{2+}$ crystals in mother liquid containing 100–200 mM MnCl$_2$ for 3–24 h. Complex crystals of hEXOG–H140A with Mg$^{2+}$ and DNA were obtained using the sitting-drop method at 17 °C using hEXOG–H140A (6–7.0 mg ml$^{-1}$), a 2.3 M ratio of DNA and 10 mM MgCl$_2$ against a well solution containing 22–28% PEG 2K MME, 0.2–0.25 M (NH$_4$)$_2$SO$_4$ and 0.1 M Na acetate pH 4.6. Crystals with Mn$^{2+}$ were obtained by soaking EXOG-H140A:Mg$^{2+}$:dsDNA complex crystals in mother liquor with 75–150 mM MnCl$_2$ for 5–24 h. All crystals were cryo-protected by stepwise increasing concentrations of glycerol and MPD in mother liquor until both reached 15%. They were then flash-frozen in liquid nitrogen. Single-wavelength anomalous diffraction data were collected using the hEXOG apo–Mn$^{2+}$ or DNA–Mn$^{2+}$ complex crystals at an energy level of 6.539 keV-the K-edge of Mn$^{2+}$ anomalous scattering. Data were processed using HKL2000 (ref. 38). The difference anomalous map calculated from the single-wavelength anomalous diffraction data shows peaks above 5-sigma at the same position as the original Mg$^{2+}$ ions, confirming their replacement with Mn$^{2+}$ (Supplementary Fig. 3b,c). This suggests that the metal ion in the original crystals contain Mg$^{2+}$ at the same locations.

**Structure determination.** The crystal structure of hEXOG:Mg$^{2+}$ was determined by molecular replacement using Phaser and *Drosophila* EndoG (PDB accession number 3ISM) as search model[39]. The missing Wing domain was built using Coot[40], and refined using Phenix[41]. Crystal structures of hEXOG–DNA complexes were determined by molecular replacement using the apo EXOG structure. MolProbity[42] was used to evaluate the final structures. Single-wavelength anomalous dispersion data were collected for apo and DNA complex containing Mn$^{2+}$. The crystal structures of hEXOG:Mn$^{2+}$ and hEXOG:DNA:Mn$^{2+}$ were

determined by molecular replacement using hEXOG:Mg$^{2+}$ and hEXOG–DNA structure as respective models. Metal locations were determined from an anomalous difference map. After refinement, Ramachandran favoured residues were ≥98% and Ramachandran outlier residues were ≤0.57% for all four structures. All structural figures were prepared using PyMol.

**Data availability.** Coordinates and structure factor of the structure reported here have been deposited into the Protein Data Bank with PDB codes: 5T40, 5T3V, 5T5C and 5T4I. All additional experimental data are available from the corresponding author on request.

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

## Acknowledgements

We thank C. Shumate for critical reading of the manuscript, Seipo Co. for artistic rendering, B. Szczesny and S. Mitra for the original clone of hEXOG. We thank W. Bujalowski for the usage of ISS PC1 photon counting spectrofluorimeter and helpful discussions. We thank staff at the beamlines 19-ID, 19-BM and 21-ID at the Advanced Photon Source, Argonne National Laboratory, and beamlines 8.2.1 and 8.2.2 at the Advanced Light Source, Lawrence Berkeley National Laboratory. M.R.S. was a J.B. Kempner Postdoctoral Fellow. A.M.G. and J.C.L. are supported by an endowment (H-0013) from the Welch Foundation. The work was supported by grants from NIH (GM 083703 and GM110591) to Y.W.Y., and an endowment from Sealy and Smith Foundation to Sealy Center for Structural Biology at UTMB.

## Author contributions

M.R.S., J.C.L. and Y.W.Y. conceived the project and designed the experiments. M.R.S., W.Y., A.M.G., M.A.W. and Y.W.Y. performed the experiments. M.R.S., I.J.M. and Y.W.Y. wrote the manuscript; all authors provided editorial suggestions and criticisms.

## Additional information

**Competing interests:** The authors declare no competing financial interests.

