## [Peer Review File · Nature Communications]

Reviewers' Comments:

Reviewer #1 (Remarks to the Author)

Szymanski et al report a crystal structure of human EXOG in its near full-length apo form, and bound to a substrate DNA duplex, along with biochemical analysis of EXOG catalytic activity. The structure of the EXOG-DNA complex highlights a role for the "wing" domain in providing specificity for interaction with duplex DNA structure, and for cutting the DNA backbone two nucleotides away from the 5' phosphorylated end. This biochemical and structural observation has an interesting connection to the biochemical attributes of the polymerase that works along with the endonuclease to repair mitochondrial DNA damage. The polymerase apparently does not synthesize new DNA efficiently on single nucleotide gaps but does much better with two-nucleotide gaps. Thus the two-nucleotide cuts of EXOG provides a product DNA that is appropriate for the mitochondrial repair polymerase. Overall, the study is a nice contribution and should appeal to the other researchers working in this area. As detailed below, the paper has several problems with presentation and conclusions drawn from some of the results. These problems can mostly be fixed by further work on the writing of the story, but some of the data is not very convincing (ITC), or is not clearly connected to the conclusions.

-page 6, line 3. "(Fig. 1a), the reaction product is a dinucleotide," This image does not have size markers to confirm that this is true.

-page 6, "hEXOG incises duplex and gapped DNA with equal efficiency (Fig. 1b), suggesting that the enzyme has higher affinity towards a 5'-P, as the gapped DNA substrate has twice the number of 5'-OH ends than the duplex." This assay does not monitor what is happening to the 5'OH ends of the DNA substrates, and the authors seem to be arguing that the excess of 5'OH ends should have lowered apparent activity through competition for protein binding, if the the protein indeed binds to 5'OH ends. There are a lot of assumption here, so it would be better to directly measure whether EXOG acts on DNA substrates lacking the 5'P, or to directly measure the affinity for DNA with 5'OH.

-Figure 1D, it is difficult to see the data at the earliest times points in this plot. An expansion of the early time points would help with the evaluation of the data.

-It is not made clear how the data in Supp. Figure 1C are supposed to report on the active fraction of EXOG for DNA binding. It is stated that "90% of monomer can bind to DNA" but the data rather seems to show that only 90% of the DNA is capable of being bound.

-ITC experiments. The data presented do not have a high level of signal to noise, and the heat of dilution is making a strong contribution to the processed data. Thus, it is hard to make strong conclusions from this data without some types of controls for what is being suggested. It seems the plot in Supp. Fig. 2A should have an x-scale in minutes, not seconds.

-SAXS. There should be more detail regarding the processing of the SAXS data. XCS is not defined anywhere that I could find. What do the Guinier plots look like for the different data sets that were merged. How well do alternate coordinate models fit the SAXS data? For example, with only one DNA molecule bound?

-EK140 buffer is not defined.

-Crystallography. The Ramachandran outliers should be defined and justified. It would be helpful to add the CC* statistic to the crystallographic table.

page 13, "Notably, however, the catalytic efficiency of the R314A mutant is ~10-fold higher than wild-type (Fig. 5b,c)" It is not clear how this rate is being calculated.

page 14, "It remodels the active site" I think it is more accurate to say that the Wing domain remodels the DNA binding site. The lack of changes in the core was highlighted several times in the manuscript.

page 15, "The mechanism of cleavage site positioning suggests that hEXOg is likely to process a damaged 5' end regardless of chemical structure, circumventing a potential blockage in the DNA repair pathway. " This needs better/more explanation. Do the authors think that hEXOg will process 5' ends with hydroxyl groups? It seems that the Arg binding to the 5'P is important, how will other chemical structures fit into the DNA binding site? Only 5'P are investigated in this study.

page 15, "this study has established the indispensable role of hEXOg in mitochondrial DNA repair." As written, this is an overstatement. I believe the authors intend to say that they have shown the precise activity of hEXOg, which makes it indispensable for mito DNA repair, as shown by others.

-The molecular weight standards for the gel filtration analysis need to be shown.

page 16, the paragraph on wing domain acting like a intrinsically disordered protein and forming protein-protein interactions is wildly speculative and should be removed. The physical basis for a well ordered and a poorly ordered DNA binding site in the hEXOg homodimer is not determined in this study.

-page 6, line 2. "The" is capitalized but should not be.

-page 7, line 3. "To determine whether the results are due to that only a half of the monomers are capable of binding to DNA" This sentence does not read well.

- The "DNA docking" label in Fig. 4c gives the impression that the DNA was docked onto the structure.

Reviewer #3 (Remarks to the Author)

Summary

The authors report the apo and DNA-bound crystal structures of human mitochondrial hEXOg. They identified a Wing domain that confers 5' exonuclease activity on duplex DNA and excises a dinucleotide, thereby creating a better substrate for gap filling synthesis by polymerase gamma. These findings are important because they identify a novel step in the mitochondrial BER pathway. I have some comments listed below.

1. For the structural data, the authors comment that the RSRZ scores are high for the DNA-bound structures, but do not address why these scores are high even in the apo structures. This needs to be addressed, as there could be errors in model building that could be fixed. Another reason why this is important is that the apo structure was used during molecular replacement to solve the DNA-bound structures that could contribute to the poor RSRZ scores in the DNA bound structures. Also, it would be useful to know what the molecular replacement statistics are.

2. The figures are not described in order in the text. For example, Figure 2b is not discussed at all and parts of figure 1 and 3 are described non-sequentially making it hard to follow along.

3. There appears to be a typo in Table 1: The first apo hEXOg complex I structure PDB is labeled as 3T40, should this be 5T40?

4. A better discussion of what domain swapping is and if it is seen with other similar proteins would greatly help the reader understand why this is important in this case.

5. On page 9, lines 189, the authors suggest "that the 5'-P is in the active site near H140". This is not apparent from any of the figures. As such a figure depicting this would be useful to the reader.

6. Also on page 9, the title "DNA binding disrupts hEXOG homodimeric structure" is misleading as the core domain structure does not change and only the wing domains undergo conformational changes.

Point-by-point responses to reviewers' comments

The reviewers' comments are in blue, and our responses are in black.

Reviewer 1

Page 6, line 3. "(Fig. 1a), the reaction product is a dinucleotide," This image does not have size markers to confirm that this is true.

We have substituted Fig. 1a with a gel that contains synthetic 20-nt and 2-nt oligonucleotides as markers.

Page 6, "hEXOg incises duplex and gapped DNA with equal efficiency (Fig. 1b), suggesting that the enzyme has higher affinity towards a 5'-P, as the gapped DNA substrate has twice the number of 5'-OH ends than the duplex." This assay does not monitor what is happening to the 5'OH ends of the DNA substrates, and the authors seem to be arguing that the excess of 5'OH ends should have lowered apparent activity through competition for protein binding, if the the protein indeed binds to 5'OH ends. There are a lot of assumption here, so it would be better to directly measure whether EXOG acts on DNA substrates lacking the 5'P, or to directly measure the affinity for DNA with 5'OH.

The reviewer correctly points out that we did not provide a direct comparison between 5'P and 5'OH ends. Some fluorescence experiments had, in fact, been done. We had not included the data because of space limitations and differences in experimental design. We now present fluorescence anisotropy experiments for hEXOg binding to 5'P and 5'OH DNA, the method is more accurate because it measures binding under equilibrium condition. We show that hEXOg binds to 5'P-DNA 3-fold more tightly than 5'OH DNA. The results are comparable to the EMSA data shown in the manuscript (now Supplementary Fig 1d, for direct comparison). Details of the new experiments are in the revised Supplementary Fig. 1c; experimental details are described on p6, line 10-14, and p21, line 3-10.

Figure 1D, it is difficult to see the data at the earliest times points in this plot. An expansion of the early time points would help with the evaluation of the data.

We expanded the initial time points as the reviewer suggested and incorporated it in Fig. 1.

It is not made clear how the data in Supp. Figure 1C are supposed to report on the active fraction of EXOG for DNA binding. It is stated that "90% of monomer can bind to DNA" but the data rather seems to show that only 90% of the DNA is capable of being bound.

It has been rephrased (p7, line 7).

ITC experiments. The data presented do not have a high level of signal to noise, and the heat of dilution is making a strong contribution to the processed data. Thus, it is hard to make strong conclusions from this data without some types of controls for what is being suggested. It seems the plot in Supp. Fig. 2A should have an x-scale in minutes, not seconds.

We thank the reviewer for drawing attention to this experiment. We made an unfortunate error in transcribing reaction conditions from a preliminary experiment rather than the experiment actually shown in Supplementary Fig. 2. Methods have now been revised to accurately describe our experimental conditions (p19, line 23, p20 lines 1-2), and the heats of dilution now correlate with the volumes of added titrant.

We agree with the Reviewer that the heat of dilution of the oligonucleotide makes a strong contribution to the processed data. However, this is an intrinsic property of the system. We had tested several factors that affect heat exchange, including temperature, pH and enzyme concentrations, to

increase the heat of binding relative to the heat of dilution. The maximum differential is achieved under the reported conditions within the solubility limits of the hEXOg-DNA complex. These conditions are reproducible: three completely independent replicate experiments yield similar binding constants difference, at 48, 55 and 58-fold, for hEXOg dimer.

We also corrected the Supplementary Fig.2a label from 'seconds' to 'minutes'. Thank you.

SAXS. There should be more detail regarding the processing of the SAXS data. XCS is not defined anywhere that I could find. What do the Guinier plots look like for the different data sets that were merged. How well do alternate coordinate models fit the SAXS data? For example, with only one DNA molecule bound?

We now include the Guinier plots for apo and DNA bound hEXOg and replaced XCS with the new figure title in Supplementary Fig. 6. We added more experimental details (p19, line 12-15). We also fitted the SAXS data to a hEXOg complexed to a single DNA and compared it to the two DNA complexes. χ^2 values are, respectively, 4.8 and 1.8 for one and two DNAs bound to a hEXOg dimer, supporting our conclusions from both the crystallographic and ITC data.

EK140 buffer is not defined.

EK140 is now defined (p18 line 10-11).

Crystallography. The Ramachandran outliers should be defined and justified. It would be helpful to add the CC* statistic to the crystallographic table.

The one Ramachandran outlier has been corrected. We included CC* statistics in Table 1.

Page 13, "Notably, however, the catalytic efficiency of the R314A mutant is ~10-fold higher than wild-type (Fig. 5b,c)" It is not clear how this rate is being calculated.

Because our reaction rate is underestimated, we revised the sentence to 'The reaction rate is at least 2-fold faster, together with 6-fold lowered binding affinity of R314A to the 2-nt product, the increased catalysis is probably caused by faster product release' (p13, line 12-15).

Page 14, "It remodels the active site" I think it is more accurate to say that the Wing domain remodels the DNA binding site. The lack of changes in the core was highlighted several times in the manuscript.

We rephrased the sentence as the reviewer suggested (p14, line 19)

Page 15, "The mechanism of cleavage site positioning suggests that hEXOg is likely to process a damaged 5' end regardless of chemical structure, circumventing a potential blockage in the DNA repair pathway. " This needs better/more explanation. Do the authors think that hEXOg will process 5' ends with hydroxyl groups? It seems that the Arg binding to the 5'P is important, how will other chemical structures fit into the DNA binding site? Only 5'P are investigated in this study.

We only intended to suggest is that EXOG should be able to process phosphate containing lesions of BER, e.g., 5'dRP (5'P-dexoyribose) and 5'P-dL (5'P-deoxyribonolactone). hEXOg should also remove 5'OH albeit at lower rate because the 5'P-R314 interaction is missing. The topic is now clarified on p15, line 3-9.

Page 15, "this study has established the indispensable role of hEXOg in mitochondrial DNA repair." As written, this is an overstatement. I believe the authors intend to say that they have shown the precise activity of hEXOg, which makes it indispensable for mito DNA repair, as shown by others.

We have rephrased the according to the reviewer's suggestion (p15 line 15-17).

The molecular weight standards for the gel filtration analysis need to be shown.

Molecular weight standards were added to Supplementary Figure 1b.

Page 16, the paragraph on wing domain acting like a intrinsically disordered protein and forming protein-protein interactions is wildly speculative and should be removed. The physical basis for a well ordered and a poorly ordered DNA binding site in the hEXOG homodimer is not determined in this study.

We agree with the reviewer, and revised the paragraph (p16, line 12-17).

Page 6, line 2. "The" is capitalized but should not be.

Corrected.

Page 7, line 3. "To determine whether the results are due to that only a half of the monomers are capable of binding to DNA" This sentence does not read well.

We have rephrased the sentence (p7, lines 5-6).

The "DNA docking" label in Fig. 4c gives the impression that the DNA was docked onto the structure.

The legend of Fig. 4c has been revised (p33, line 7-8).

Reviewer 2

1. For the structural data, the authors comment that the RSRZ scores are high for the DNA-bound structures, but do not address why these scores are high even in the apo structures. This needs to be addressed, as there could be errors in model building that could be fixed. Another reason why this is important is that the apo structure was used during molecular replacement to solve the DNA-bound structures that could contribute to the poor RSRZ scores in the DNA bound structures. Also, it would be useful to know what the molecular replacement statistics are.

The RSRZ score for the apo structures is 4.5 %, much lower than the 11.5% for the DNA complex structure. We examined ten recent protein structures determined at the comparable resolution (1.5-2Å) (pdb accession numbers: pdb accession codes: 5B41, 5B5Y, 5DAL, 5EY3, 5E41, 5EXG, 5F1F, 5EY6, 5F2I, and 5F2M), RSRZ scores range from 3.3-9.6%. The RSRZ score for apo hEXOG is well within the range.

PDB staff suggested to set occupancy of residues with high RSRZ to zero.

Other suggestions from CCP4 bulletin board are to distort the geometry of these residues in exchange of low RSRZ scores, or to remove hydrogen atoms before running validation. These operations indeed decrease RSRZ score, but at the expense of eliminating residues with weaker electron density or resulting in incorrect configurations. Removing hydrogens in the structure during refinement will result in increased steric clashes. These approaches are currently contentious in the crystallographic community.

The molecular replacement statistics were added to Table 1.

2. The figures are not described in order in the text. For example, Figure 2b is not discussed at all and parts of figure 1 and 3 are described non-sequentially making it hard to follow along.

Fig 2b is now referred to. All figures are referred sequentially. Perhaps the Reviewer intended to critique supplementary figure 1 and 3. Supplementary figure 1 is referred multiple times throughout the ms, may appear to be out of order.

3. There appears to be a typo in Table 1: The first apo hEXOG complex I structure PDB is labeled as 3T40, should this be 5T40?

Corrected.

4. A better discussion of what domain swapping is and if it is seen with other similar proteins would greatly help the reader understand why this is important in this case.

We added a descriptive sentence explaining domain swapping (p8, line 5-7). This is the first structure of an exonuclease in the $\beta\beta\alpha$ -Me family nucleases; the other family members are endonucleases without domain swapping. We concluded from the biochemical and structural analyses that stable dimer is more important for EXOG than for EndoG (p8 line 10-13).

5. On page 9, lines 189, the authors suggest “that the 5'-P is in the active site near H140”. This is not apparent from any of the figures. As such a figure depicting this would be useful to the reader.

The sentence is revised to ‘the 5'-P is proximal, and the 5'-OH is distal’ to the active site (p9, line 14-16).

6. Also on page 9, the title “DNA binding disrupts hEXOG homodimeric structure” is misleading as the core domain structure does not change and only the wing domains undergo conformational changes.

The subheading has been revised (p9, line 18).

Reviewers' Comments:

Reviewer #1 (Remarks to the Author)

Szymanski et al have addressed the key concerns raised in the review of their manuscript. The Guinier plots added to the Supplement could use a better description of what the blue line refers to (presumably residuals of the fit), and over what range of the data the Guinier was calculated. It is clear that not all points are included in the residual calculation, since there are some clear deviations that do not show up in the residual plot. Also, there also appears to be a duplication of the 2.0 mg/ml concentration for the "hEXOG-DNA-compex (sic)" (they have misspelled complex in the right heading of panel d.

Point-by-point responses to reviewers' comments

The reviewer's comments are in blue, and our responses are in black.

Reviewer #1:

Szymanski et al have addressed the key concerns raised in the review of their manuscript. The Guinier plots added to the Supplement could use a better description of what the blue line refers to (presumably residuals of the fit), and over what range of the data the Guinier was calculated. It is clear that not all points are included in the residual calculation, since there are some clear deviations that do not show up in the residual plot. Also, there also appears to be a duplication of the 2.0 mg/ml concentration for the "hEXOG-DNA-compex (sic)" (they have misspelled complex in the right heading of panel d.

The blue line represents normalized residuals (= residual/error). The Guinier plots were calculated over the range from 0.0122 to 1.3/Rg (\AA^{-1}) for all concentrations of hEXOG-DNA-complex with exception of the one at 2.0mg/ml that was from 0.0162 to 1.3/Rg (\AA^{-1}). We added this information to the legend of Supplementary Figure 6.

We corrected the typo and removed the duplication.